# The Importance of Interprofessional Collaboration (IPC) Guidelines in Stunting Management in Indonesia: A Systematic Review

**DOI:** 10.3390/healthcare12222226

**Published:** 2024-11-07

**Authors:** Rachmat Sentika, Trisna Setiawan, Deborah Johana Rattu, Irma Yunita, Bertri Maulidya Masita, Ray Wagiu Basrowi

**Affiliations:** 1Indonesian Community Health Center Acceleration, APKESMI, Daerah Khusus Ibukota Jakarta 10160, Indonesia; rsentikaspa@gmail.com (R.S.); trisnaku.medika@gmail.com (T.S.); fayruzkusnadi2@gmail.com (K.); deborahjrattu@gmail.com (D.J.R.); dr.irmayunita@gmail.com (I.Y.); 2Medical and Scientific Affairs, Specialized Nutrition, Jakarta 12940, Indonesia; bertri.masita@danone.com; 3Occupational Medicine Study Program, Department of Community Medicine, Faculty of Medicine, Universitas Indonesia, Jakarta 10430, Indonesia

**Keywords:** stunting, systematic review, interprofessional collaboration, prevention, primary health center

## Abstract

**Background:** Indonesia’s stunting prevention programs have shown limited effectiveness and scalability. Interprofessional collaboration (IPC) is crucial for preventing and treating pediatric stunting. This study aimed to investigate the roles of primary health centers and IPC in addressing stunting in Indonesia. **Methods:** A systematic review was conducted, searching PubMed, EMBASE, Web of Science, ProQuest, and Google Scholar for studies up to November 2023. Two reviewers independently screened and included full-text articles that examined IPC and stunting, focusing on prevalence, policy implications, healthcare roles, community roles, preventive measures, and challenges. **Results:** The review included 52 articles. IPC was found to be critical in reducing stunting prevalence. Training improved the knowledge of health cadres, aiding in the early detection and prevention of stunting. The studies emphasized IPC’s positive impact on stunting reduction across various regions. Policymakers are encouraged to adopt a comprehensive strategy involving collaboration, financial support, and effective program implementation. **Conclusions:** This study highlights IPC as essential in reducing stunting in Indonesia. Integrating stunting management into primary healthcare is recommended, with a call for developing guidelines to standardize and optimize the approach to this public health issue.

## 1. Introduction

The protracted prevalence of stunting in Indonesia for numerous decades has been attributed to nutritional insufficiencies and recurrent early childhood infections [1]. Despite a decline from previous years, data from the 2022 Indonesia Nutritional Status Survey indicate that the prevalence of stunting remains at 21.6% [2]. According to World Health Organization (WHO) standards, this figure still constitutes a public health problem (>20%) [3]. Studies demonstrate that stunting prevention is more effective than treatment through interventions such as integrated nutrition interventions and addressing sociodemographic factors, nutrition, and environmental sanitation [4,5]. The Indonesian government has also made efforts to accelerate the reduction of stunting through specific, sensitive, convergent, holistic, integrative, and quality interventions, particularly at community health centers [6]. Taken together, stunting continues to be a challenge in Indonesia, emphasizing the need for strategic measures to reduce its prevalence.

Various stunting prevention programs have been implemented in Indonesia; however, their effectiveness remains suboptimal and lacks adequate scalability. A study elucidated the pivotal role of accurate nutritional status monitoring, tiered referral protocols, and nutritional interventions in stunting management [7]. In a prior investigation, the emphasis on the administration of protein-rich supplementary foods over a 90-day period yielded a notable outcome, manifesting as a 62.1% improvement in the nutritional status of young children. Furthermore, in cases of severe malnutrition, administering F-75 (a therapeutic milk formula containing 75 kcal/100 mL used in initial phases of treatment) for 3 days and F-100 (a more energy-dense therapeutic milk formula containing 100 kcal/100 mL used in the rehabilitation phase) for 11 days proved successful in enhancing the nutritional status of 46.2% of young children [8]. The study underscored the critical role in addressing cases at risk of growth failure, undernutrition, and severe malnutrition as predisposing factors or stages in the risk of stunting among young children [9]. Diagnosis and intervention for children with malnutrition require medical professionals at primary health centers with expertise in medicine, nutrition, midwifery, and nursing [10]. These findings underscore the pivotal role of primary health centers in addressing stunting in Indonesia.

Healthcare professionals wield a pivotal role in the mitigation of stunting, as evidenced by collaborative endeavors demonstrating success in reshaping maternal attitudes and caregiving behaviors. Proficiency in stunting-related knowledge stands as a crucial component among healthcare practitioners, essential for effective prevention and reduction strategies [11]. In-depth studies emphasize the profound impact of healthcare professional involvement, empowering mothers through the fortification of knowledge and skills in self-management for optimal pregnancy preparation and child development [12]. Empirical evidence highlights that improving accessibility and elevating educational standards in nutritional science, particularly through healthcare services, enhances maternal knowledge and elevates child feeding practices, collectively contributing to a significant reduction in stunting prevalence [13]. Those reports demonstrate that interprofessional collaboration emerges as a central imperative in the scientific approach to preventing and treating pediatric stunting in Indonesia.

Epidemiological studies have provided compelling evidence underscoring the pivotal role of primary health centers and interprofessional collaborations in the prevention and eradication of stunting. Nevertheless, a systematic review examining the connection between the functions of primary health centers and interprofessional collaborations in addressing stunting is still lacking. The objective of this study was to investigate the roles of primary health centers and interprofessional collaborations in association with stunting, along with exploring potential measures within the context of Indonesia.

## 2. Materials and Methods

### 2.1. Data Sources and Search Strategy

This systematic review and meta-analysis adhered to the guidelines outlined in the Preferred Reporting Items for Systematic Reviews and Meta-Analyses (PRISMA) protocols [14]. The research plan was preregistered on PROSPERO (CRD42024607995) prior to the commencement of the study. The comprehensive checklist and study protocol can be found in the Appendix A. A systematic search of pertinent databases, including PubMed, EMBASE, Web of Science, ProQuest, and Google Scholar, was conducted up until 15 November 2023. Our investigation imposed no constraints on publication date; however, it was confined to articles published in the English language. The search strategy was centered on identifying studies that explored the association between interprofessional collaboration and stunting, with a focus on estimating prevalence, policy implications, healthcare roles, community roles, preventive measures, and challenges. A combination of the following terms was used to search keywords, titles, and abstracts: “stunting”, “stunted”, “growth disorder”, “malnutrition”, “retardation”, “failure to thrive”, “growth restriction”, “growth retardation”, and “growth impairment”.

### 2.2. Study Selection

The outcomes of the electronic searches were imported into the EndNote X20 bibliographic software for further analysis. Inclusion criteria for the study on pediatric stunting and interprofessional collaboration were as follows: (1) consideration of outcomes related to pediatric stunting in the context of interprofessional collaboration; (2) studies providing estimates for relevant outcomes, such as prevalence rates, intervention effectiveness, healthcare utilization, mortality, symptoms, and diagnoses associated with pediatric stunting; (3) adherence to the diagnostic criteria for pediatric stunting as defined by recognized standards or guidelines; and (4) publication in a scientific journal without any restrictions on the publication date. Exclusion criteria encompassed studies that (1) were not full-text scientific articles; (2) were review articles, letters, or commentaries; (3) that did not present quantitative estimates; (4) that were not written in English; (5) in which the study location was not in Indonesia, and (6) that were re-evaluations or duplicates of prior studies.

### 2.3. Data Extraction

The data we extracted consisted of general study characteristics including (1) first author, (2) year of publication, (3) study design, and (4) title. We used the Joanna Briggs Institute’s critical appraisal checklist to assess the risk of bias in the selected study. Two independent reviewers conducted individual assessments of each included study, and any disagreements were resolved through collaborative team discussions to reach a consensus. The risk of bias assessment is discussed in more detail in Table 1.

**Table 1 healthcare-12-02226-t001:** Quality assessment and risk of bias of included studies.

No.	Study (Publication Year)	Were the Criteria for Inclusion in the Sample Clearly Defined?	Were the Study Subjects and the Setting Described in Detail?	Was the Exposure Measured in a Valid and Reliable Way?	Were Objective Standard Criteria Used for Measurement of the Condition?	Were Confound-Ing Factors Identified?	Were Strategies to Deal with Confounding Factors Stated?	Were the Outcomes Measured in a Valid and Reliable Way?	Was Appropriate Statistical Analysis Used?	Total
1	Absori et al., 2022 [1]	Yes	Yes	Yes	Yes	No	No	Yes	Yes	6
2	Adi et al., 2019 [2]	Yes	Yes	Yes	Yes	No	No	Yes	Yes	6
3	Ain et al., 2023 [3]	Yes	Yes	Yes	Yes	No	Yes	Yes	Yes	7
4	Akhmadi et al., 2021 [4]	Yes	Yes	Yes	Yes	Yes	Yes	Yes	Yes	8
5	Anastasia et al., 2023 [5]	Yes	Yes	Yes	Yes	Yes	Yes	Yes	Yes	8
6	Astikasari et al., 2023 [6]	Yes	Yes	Yes	Yes	No	No	Yes	Yes	6
7	Astuti et al., 2021 [7]	Yes	Yes	Yes	Yes	Yes	No	Yes	Yes	7
8	Azis et al., 2023 [8]	Yes	Yes	Yes	Yes	No	No	Yes	Yes	6
9	Barber et al., 2019 [9]	Yes	Yes	Yes	Yes	Yes	Yes	Yes	Yes	8
10	Daniel et al., 2023 [10]	Yes	Yes	Yes	Yes	No	No	Yes	Yes	6
11	Erfina et al., 2023 [11]	Yes	Yes	Yes	Yes	No	Yes	Yes	Yes	7
12	Estiwidani et al., 2021 [12]	Yes	Yes	Yes	Yes	Yes	Yes	Yes	Yes	8
13	Fikawati et al., 2021 [13]	Yes	Yes	Yes	Yes	Yes	Yes	Yes	Yes	8
14	Gani et al., 2021 [14]	Yes	Yes	Yes	Yes	Yes	No	No	Yes	6
15	Hadi et al., 2021 [15]	Yes	Yes	Yes	Yes	Yes	Yes	Yes	Yes	8
16	Herawati et al., 2022 [16]	Yes	Yes	Yes	Yes	No	No	Yes	Yes	6
17	Huriah et al., 2023 [17]	Yes	Yes	Yes	Yes	No	Yes	Yes	Yes	7
18	Ipa et al., 2020 [18]	Yes	Yes	Yes	Yes	No	No	Yes	Yes	6
19	Julia et al., 2004 [19]	Yes	Yes	Yes	Yes	No	No	Yes	Yes	6
20	Khasanah et al., 2022 [20]	Yes	Yes	Yes	Yes	No	No	Yes	Yes	6
21	Laksono et al., 2022 [21]	Yes	Yes	Yes	Yes	No	No	Yes	Yes	6
22	Latupeirissa et al., 2020 [22]	Yes	Yes	Yes	Yes	No	Yes	Yes	Yes	7
23	Maloho et al., 2023 [23]	Yes	Yes	Yes	Yes	Yes	No	No	Yes	6
24	Mairo et al., 2020 [24]	Yes	Yes	Yes	Yes	No	No	Yes	Yes	6
25	Mairo et al., 2022 [25]	Yes	Yes	Yes	Yes	No	No	Yes	Yes	6
26	Maulina et al., 2022 [26]	Yes	Yes	Yes	Yes	No	No	Yes	Yes	6
27	Mayfitriana et al., 2022 [27]	Yes	Yes	Yes	Yes	No	No	Yes	Yes	6
28	Maryati et al., 2022 [28]	Yes	Yes	Yes	Yes	Yes	Yes	Yes	Yes	8
29	Mediani et al., 2022 [29]	Yes	Yes	Yes	Yes	No	No	Yes	Yes	6
30	Mindarsih et al., 2023 [30]	Yes	Yes	Yes	Yes	No	No	Yes	Yes	6
31	Muliadi et al., 2023 [31]	Yes	Yes	Yes	Yes	No	No	Yes	Yes	6
32	Mulyani et al., 2022 [32]	Yes	Yes	Yes	Yes	No	No	Yes	Yes	6
33	Mutiarasari et al., 2021 [33]	Yes	Yes	Yes	Yes	Yes	No	No	Yes	6
34	Nawan et al., 2020 [34]	Yes	Yes	Yes	Yes	Yes	Yes	Yes	Yes	8
35	Nazri et al., 2016 [35]	Yes	Yes	Yes	Yes	Yes	No	Yes	Yes	7
36	Neherta et al., 2021 [36]	Yes	Yes	Yes	Yes	No	Yes	Yes	Yes	7
37	Nizaruddin et al., 2023 [37]	Yes	Yes	Yes	Yes	Yes	Yes	Yes	Yes	8
38	Novitasari et al., 2020 [38]	Yes	Yes	Yes	Yes	Yes	No	No	Yes	6
39	Oktaviana et al., 2022 [39]	Yes	Yes	Yes	Yes	No	Yes	Yes	Yes	7
40	Permatasari et al., 2019 [40]	Yes	Yes	Yes	Yes	No	No	Yes	Yes	6
41	Prastiwi et al., 2020 [41]	Yes	Yes	Yes	Yes	No	No	Yes	Yes	6
42	Ramli et al., 2009 [42]	Yes	Yes	Yes	Yes	Yes	Yes	Yes	Yes	8
43	Rizal et al., 2022 [43]	Yes	Yes	Yes	Yes	No	No	Yes	Yes	6
44	Saadah et al., 2022 [44]	Yes	Yes	Yes	Yes	No	No	Yes	Yes	6
45	Sinaga et al., 2021 [45]	Yes	Yes	Yes	Yes	No	No	Yes	Yes	6
46	Sukmawati et al., 2021 [46]	Yes	Yes	Yes	Yes	No	No	Yes	Yes	6
47	Suratri et al., 2023 [47]	Yes	Yes	Yes	Yes	No	No	Yes	Yes	6
48	Suryani et al., 2022 [48]	Yes	Yes	Yes	Yes	Yes	Yes	Yes	Yes	8
49	Syafrawati et al., 2022 [49]	Yes	Yes	Yes	Yes	No	No	Yes	Yes	6
50	Susanto et al.,2021 [50]	Yes	Yes	Yes	Yes	Yes	Yes	Yes	Yes	7
51	Titaley et al., 2019 [51]	Yes	Yes	Yes	Yes	Yes	Yes	Yes	Yes	8
52	Widhiastuti et al., 2021 [52]	Yes	Yes	Yes	Yes	No	No	Yes	Yes	6

### 2.4. Data Synthesis and Analysis

The analysis followed a systematic, multi-step approach guided by established protocols for mixed-methods systematic reviews. Narrative synthesis was employed as the primary analytical method due to the heterogeneity in study designs and outcome measures across included studies. Two independent reviewers conducted a thematic analysis of the extracted data, beginning with the initial coding of key findings and concepts, followed by grouping related codes into broader themes, identifying patterns and relationships between themes, and synthesizing findings within and across thematic areas. The analysis focused on six key domains: prevalence data and trends, policy frameworks and implementation, healthcare worker roles and capabilities, community engagement mechanisms, prevention program characteristics, and implementation challenges and barriers. Regular team discussions were held to resolve coding discrepancies, refine thematic categories, draw connections between findings, and validate interpretations. Quality assessment scores were considered when weighing evidence strength and making recommendations. The synthesis prioritized identifying actionable insights for policy and practice while acknowledging the methodological limitations of the included studies.

## 3. Results

### 3.1. Study Characteristics

A database search resulted in 2672 studies, and 408 studies were excluded due to duplicate publications (Figure 1). One hundred and thirty-eight eligible studies were collected for the evaluation of their full text. We excluded 12 conference abstracts, reviews, and protocols, and 74 studies did not report outcomes related to stunting management in healthcare facilities. Finally, 52 articles were included in the qualitative synthesis.

Out of the 52 included studies, the outcome measurements primarily focused on the prevalence of stunting (ten studies), the healthcare role (thirteen studies), community empowerment/the role of Posyandu cadres (six studies), stunting prevention policy/management (eight studies), stunting prevention program (fourteen studies), and the challenges of stunting prevention (three studies). Characteristics of the included studies are presented in Table 2.

### 3.2. Prevalence of Stunting in Indonesia

Ten studies employed a cross-sectional design to investigate the prevalence of stunting. Four studies calculated the prevalence using anthropometric data [16,17,18,19,20], one study used Indonesia Basic Health Research (RISKESDAS) [21], one study used Indonesian Basic Health Surveys [22], one study utilized Indonesian Demographic Survey (IDHS) [23], one study used a family-based survey [24], and one study used Indonesia Nutritional Status Monitoring Survey [25]. In the North Maluku province, based on WHO child growth standards, the prevalence of stunting and severe stunting among children aged 0–23 months was 29% and 14.1%, respectively. For children aged 0–59 months, the prevalence rate was 38.4% and 18.4%. According to the Indonesia Basic Health Survey, the prevalence of stunting in 2018 was 20.1% and 20.9% for children <12 months, 19.6% and 20.1% for 12–23 months, and 60.3% and 59.0% for 24–59 months in the South Sulawesi and West Sulawesi provinces. In terms of gender, the prevalence of stunting was 61.1% in males and 38.9% in females aged 25–30 months in central Jakarta. Based on residence, the prevalence of stunting among children under 2 years old was 22.6% in urban areas and 77.4% in rural areas. Meanwhile, the prevalence in rural, poor urban, and non-poor urban areas was 28.2%, 17.5%, and 11.8%, respectively, among children aged 6 years to 7.9 years old. The study of stunting in 10 villages of central Sulawesi showed that the prevalence of stunting was 41.1% and 38.9% at the baseline and end line in under-five-year-old children. The prevalence decreased after one year of the intervention program.

### 3.3. Stunting Policy in Indonesia

Eight studies explored existing policies, including successful policies on stunting prevention [26,27,28,29,30,31,32,33]. Using a qualitative approach, one study reported that there are some policies that may reduce stunting prevalence, including the creation of regulations focusing on village authorities to eliminate stunting. This involves community empowerment through cadres; generating a stunting data record, stunting assessment, and promotion; and the evaluation of stunting programs. Another study suggested that determining the risk factors of stunting is important in order to create a prevention policy. One study reported a successful project called the District Stunting Reduction Assistance (DSRA) policy, which focuses on nutrition-specific and nutrition-sensitive interventions. However, it is only applicable in district areas, not in sub-district areas due to capacity barriers and commitments. Another study reported eight important stages that contribute to optimal stunting prevention, including mapping and analyzing the program, formulating an activity plan, engaging in discourses, issuing a regent’s decree on village authority, developing cadres, managing a stunting data management system, conducting assessments and publications, and holding annual performance reviews. One study stated that an education program among mothers was successful in reducing stunting, and a training program can be used to improve healthcare policies. Community participation and cooperation are important factors in stunting prevention programs. Another study reported about policies that have been applied in the Surabaya region, namely exclusive breastfeeding (Government Regulation No. 33/2012), the national movement for nutrition in the first thousand days of life or 1000 HPK (Presidential Regulation No. 42/2013), and collaboration with the Regional Development Planning Agency (BAPPEDA) at the provincial level in terms of national planning on stunting prevention. One study reported that the role of health workers in Posyandu is important in reducing stunting prevalence.

### 3.4. Healthcare Role in Stunting Implementation Program

There are thirteen studies that measured the role of healthcare workers regarding stunting prevention [34,35,36,37,38,39,40,41,42,43,44,45]. One study reported that nursing intervention is needed to improve the knowledge of adolescent mothers during pregnancy with specific guidelines. Another study reported that healthcare staff lack the ability to implement stunting programs. Meanwhile, another study emphasized that adequate knowledge of healthcare staff about exclusive breastfeeding and infant and young child feeding (IYCF) is an important factor that is useful for practical collaboration among inter-professional healthcare staff to implement stunting programs. One study reported that inter-professional collaboration practices have a significant contribution to the first 1000 days of life in terms of tackling stunting. Improving the number of healthcare staff, including nurses and medical doctors, contributes to health quality, particularly in addressing stunting status. Another study reported that a lack of knowledge among health officers resulted in unsatisfactory stunting prevention. One study suggested that strengthening collaboration with the community by midwives is one impactful factor that contributes to minimizing the prevalence of stunting. Another study recommended basic skills, such as being a communicator and facilitator, to support their role in stunting prevention, particularly in the first 1000 days of life. Intensive promotion regarding the risk factors of stunting should be provided by nutritionists in public health centers. One study reported that dentists have excellent knowledge regarding stunting prevention programs, and they should be involved in this program. Good collaboration between nurses and health cadres significantly contributes to stunting prevention. One study mentioned that the service quality from public health centers would affect the motivation of pregnant women in stunting programs.

### 3.5. The Role of Community Empowerment

There are six studies that reported how the community participates in stunting programs through Posyandu [46,47,48,49,50]. One study reported that Posyandu cadres involved in training programs for the first 1000 days of life could contribute to applying basic procedures for observing stunting in children. Another study suggested that collaboration between cadres and healthcare staff could improve the health status of children in West Java. Health cadres also play an important role in promoting nutrition intake and feeding behavior. Prevention strategies through cadre participation could have a positive impact on the number of stunting cases.

### 3.6. Stunting Prevention Program in Indonesia

In this inclusive study, there are fourteen studies reporting on established programs regarding stunting in Indonesia [51,52,53,54,55,56,57,58,59,60,61,62,63,64]. One study reported that an empowerment program among pregnant women leads to a better understanding of stunting prevention programs. Another study mentioned that good sanitation, especially the quality of drinking water, is one of the programs that could reduce the risk of stunting. One study stated that a nutrition program is a primary factor in reducing stunting and low birth weight, which contributes to stunting. A mother who practices good self-care leads to positive behavior in stunting programs, including feeding, sanitation, and seeking health services. One study suggested that a mental health program significantly contributes to stunting elimination. Another study reported that participation in health insurance programs and counseling during pregnancy were sensitive programs with an impact on stunting prevalence in children under five years old. Exclusive breastfeeding is one of the effective programs to reduce stunting, particularly in low-income countries. One study suggested that the early prevention of stunting could target female adolescents.

### 3.7. Implementation Challenges Regarding Stunting Programs

Three of the included studies discovered some challenges regarding stunting implementation in Indonesia [65,66,67]. One study highlighted the major factor affecting the success of the stunting program, which is collaboration between stakeholders and budgetary support. Another study reported that the program was not applicable at any locus level due to a lack of financial support, insufficient expertise, and an unsupervised program. One study stated that there were limited facilities and non-standardized tools to support the stunting observation program.

## 4. Discussion

A total of 52 studies were included in this systematic review of interprofessional collaboration in primary healthcare. The majority of the studies covered the implemented program and the role of healthcare in tackling stunting in Indonesia. Additionally, this study also reported the positive contribution of the community to the stunting prevention program, while other evidence covered the challenges of the stunting program.

Our findings were consistent with the established program called the first 1000 days of life, which involved a comprehensive strategy aimed at fostering the health and development of both mothers and children throughout the crucial initial thousand days of life. The varied policies, including regional initiatives and collaborations, highlighted the need for tailored approaches based on the unique challenges faced in different areas. A previous study suggested that a continuous collaboration between policy development and research was important to enhance the current program and achieve the goal of eradicating stunting [68]. Another policy has been incorporated into the development planning of the Indonesian government by increasing the number of healthcare staff [69]. The Coordinating Minister for Human Development and Culture of Indonesia established a national strategy for the acceleration of stunting reduction efforts at the district level [70]. Our findings suggest that the integration of policy adjustment and health resources could achieve the national goal of stunting prevention.

This study also highlighted the crucial role of healthcare workers in stunting prevention. The challenges include a lack of implementation ability among healthcare staff and insufficient knowledge. Inter-professional collaboration and the involvement of various healthcare professionals such as dentists, nutritionists, nurses, and midwives are highlighted as effective strategies, especially during the critical first 1000 days of life. A previous study stated that sufficient knowledge of diagnosis, treatment, and monitoring of healthcare staff are important in stunting prevention [71]. Another study reported that health workers encounter difficulties in accurately measuring and inputting results into the integrated application, leading to limitations in generating qualified reports. These challenges are compounded by constraints related to anthropometric tools in Posyandu and Puskesmas [72]. Taken together, improving knowledge and collaboration among healthcare staff is a comprehensive method to address the practical aspects of stunting prevention.

Community engagement has a significant effect on the reduction of stunting prevalence, particularly in Posyandu. This study observed a positive outcome resulting from cadre training, collaborative efforts, and active participation in promoting nutrition and sanitation. Previous evidence suggested that the sustained collaboration between healthcare practitioners and health cadres is essential regarding monitoring and evaluation across stunting programs, including the administration of supplementation, feeding behavior, and the measurement and assessment of anthropometric outcomes [73]. Another study concluded that cadres showed significantly improved knowledge after stunting training, which contributed to sustainable early detection and prevention of stunting [74]. The short training consistently improves the knowledge of cadres about child growth monitoring, child development monitoring, and feeding young children [75]. Collectively, adequately trained cadres play a vital role in identifying stunting at an early stage.

The synthesis of these findings also emphasizes the need for holistic strategies encompassing collaboration, financial support, program applicability, expertise, supervision, and infrastructure to overcome these hurdles. Policymakers and program implementers should consider these challenges to enhance the overall effectiveness and impact of stunting prevention initiatives in Indonesia. Similar to our findings, a previous review summarized that the challenges to accelerating stunting prevention were inadequate communication and coordination among stakeholders, insufficient resources, limited support from local governments, and disparities in Standard Operational Procedure availability at the regional level [76]. Another previous study similarly found that the major challenges regarding the implementation of nutrition prevention were related to planning, budgeting, implementation, monitoring, and evaluation [77]. Meanwhile, according to the deputy of policy support for human development and equality in the office of the vice president in the Republic of Indonesia, the challenge was to align programs and initiatives funded by various government levels, ensuring coordination among ministries and agencies to prevent duplication and stay focused on the target [78]. Taken together, policymakers need to consider a comprehensive strategy across government levels, focusing on collaboration, financial support, and effective program implementation.

This systematic review revealed several crucial guidelines for strengthening stunting prevention efforts. A comprehensive interprofessional collaboration framework emerged and was essential, requiring clear roles and responsibilities across healthcare disciplines, standardized communication protocols, integrated care pathways, and regular team meetings. The community engagement strategy highlighted the importance of investing in comprehensive training programs for community health workers, building sustainable partnerships between healthcare facilities and community organizations, developing culturally appropriate education materials, and maintaining consistent community presence through regular outreach activities. Program implementation requirements emphasized securing adequate and sustained funding mechanisms, ensuring the availability of standardized measurement tools, establishing clear monitoring frameworks, and developing robust data management systems. Policy recommendations focused on aligning programs across different government levels to prevent duplication, creating clear accountability mechanisms, establishing sustainable funding streams, and developing standardized implementation guidelines. Capacity-building priorities included regular training updates for healthcare workers, standardized certification programs for community health workers, the development of practical implementation tools, and the establishment of mentoring systems. These guidelines provide a framework for strengthening stunting prevention efforts while acknowledging the need for local adaptation based on specific contexts and resources.

There were some limitations of this study. Firstly, the heterogeneity in study methodologies and data collection techniques across the reviewed literature may introduce variations in the findings. The generalizability of our findings might be limited by the geographic and cultural diversity within Indonesia, emphasizing the need for caution in applying our conclusions universally. Moreover, the dynamic nature of healthcare systems and policies may have evolved since the included studies, affecting the relevance of our findings to the current context. Our review prioritized the English language and indexed publications to facilitate international accessibility and standardized quality assessment. Given Indonesia’s extensive research activities in stunting prevention, future reviews may benefit from including local language publications and non-indexed literature to capture additional regional insights

## 5. Conclusions

This systematic review underscores the pivotal role of interprofessional collaboration as a significant determinant in mitigating stunting prevalence in Indonesia. The evidence supports several key conclusions regarding effective stunting prevention, which requires coordinated action across healthcare disciplines, community organizations, and government levels. Investment in community health worker training and support systems emerged and were essential for program success, alongside the need for standardized protocols and guidelines to ensure consistent implementation across different settings. Sustainable funding mechanisms and political commitment proved crucial for long-term program success, complemented by regular monitoring and evaluation systems to track progress and adjust interventions as needed. We recommend the development and implementation of comprehensive interprofessional collaboration guidelines to standardize and optimize stunting prevention approaches. Future research should focus on evaluating the effectiveness of these guidelines in various contexts and identifying strategies for sustainable implementation at scale.

## Figures and Tables

**Figure 1 healthcare-12-02226-f001:**
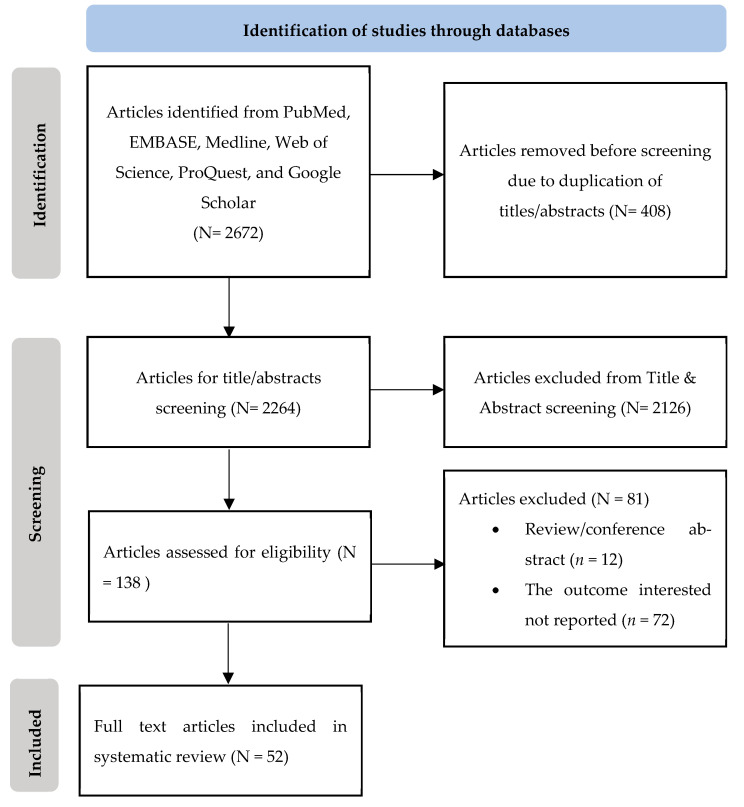
Flowchart of the study.

**Table 2 healthcare-12-02226-t002:** Characteristics of studies included in systematic review.

No	Author	Title	Study Design
1	Absori et al., 2022 [1]	Public Health-Based Policy on Stunting Prevention in Pati Regency, Central Java, Indonesia	Empirical (non-doctrinal) legal research
2	Adi et al., 2019 [2]	The Correlation between Regulation Understanding by Inter-Professional first 1000 days of Life Health Workers and the Acceleration of Toddler Stunting Prevention	Case-control study
3	Ain et al., 2023 [3]	Dynamic Self-Determination of Self-Care and Positive Deviance Model for Stunting Prevention in Indonesia	Quasi experimental study
4	Akhmadi et al., 2021 [4]	Effect of care for child development training on cadres’ knowledge, attitude, and efficacy in Yogyakarta, Indonesia	Quasi-experimental research
5	Anastasia et al., 2023 [5]	Determinants of stunting in children under five years old in South Sulawesi and West Sulawesi Province: 2013 and 2018 Indonesian Basic Health Survey	Cross-sectional study
6	Astikasari et al., 2023 [6]	Posyandu Cadres on Capacity Building: Prevent Stunting by Improving Nutrition During The First 1000 Days Of Life	Survey study
7	Astuti et al., 2021 [7]	The Effectiveness of the Interprofessional Collaboration (IPC) Program on The Attitude of Mothers and Health Cadres on Stunting at Puskesmas Karanganom Klaten Central Java Republic of Indonesia	Quasi-experimental
8	Azis et al., 2023 [8]	Analysis of Policy Implementation of The First 1000 Days of Life Program in Overcoming Stunting in Maros District	Qualitative panel research method
9	Barber et al., 2019 [9]	Health workers, quality of care, and child health: Simulating the relationships between increases in health staffing and child length	Survey study
10	Daniel et al., 2023 [10]	Interactions of Factors Related to the Stunting Reduction Program in Indonesia: A Case Study in Ende District	Case-study
11	Erfina et al., 2023 [11]	Development and evaluation of nursing intervention in preventing stunting in children of adolescent mothers: A mixed-methods research protocol	Mixed-methods explanatory sequential (qualitative phenomenology, interpretative and quantitative)
12	Estiwidani et al., 2021 [12]	Interprofessional collaborative practice is an effort to increase behavior prevention of stunting in families with the first 1000 days of life	Randomized controlled trial
13	Fikawati et al., 2021 [13]	Energy and protein intakes are associated with stunting among preschool children in Central Jakarta, Indonesia: a case-control study	Cross-sectional study
14	Gani et al., 2021 [14]	The effect of convergent action on reducing stunting prevalence in under-five children in Banggai District, Central Sulawesi, Indonesia	Evaluation study
15	Hadi et al., 2021 [15]	Exclusive Breastfeeding Protects Young Children from Stunting in a Low-Income Population: A Study from Eastern Indonesia	Cross-sectional study
16	Herawati et al., 2022 [16]	Implementation Outcomes of National Convergence Action Policy to Accelerate Stunting Prevention and Reduction at the Local Level in Indonesia: A Qualitative Study	Qualitative study
17	Huriah et al., 2023 [17]	The Determinant Factors of Stunting Among Children in Urban Slums Area, Yogyakarta, Indonesia	Case-control study
18	Ipa et al., 2020 [18]	Breast Feeding Practice Prevention for Nutritional Stunting of Children In Buginese Ethnicity	Cross-sectional study
19	Julia et al., 2004 [19]	Influence of socioeconomic status on the prevalence of stunted growth and obesity in prepubertal Indonesian children	Cross-sectional study
20	Khasanah et al., 2022 [20]	The Effect of Sensitive Interventions on Stunting Reduction Efforts	Cross-sectional study
21	Laksono et al., 2022 [21]	Stunting among children under two years in Indonesia: Does maternal education matter?	Cross-sectional study
22	Latupeirissa et al., 2020 [22]	Analysis Risk Factors of Stunting Incidence on Toddlers in the Working Area of Porto Haria Public Health Center	Analytical observational study
23	Maloho et al., 2023 [23]	Stunting Management Policies in Improving Public Health Levels in North Bolaang Mongondow Regency	Qualitative study
24	Mairo et al., 2020 [24]	Policy Study and Stunting Prevention in Surabaya	Mix method (qualitative and quantitative)
25	Mairo et al., 2022 [25]	Exploration of Stunting Events as an Effort to Prevent Stunting in Bangkalan Regency	Exploratory qualitative study
26	Maulina et al., 2022 [26]	Prevalence and predictor stunting, wasting and underweight in Timor Leste children under five years: An analysis of DHS data in 2016	Cross-sectional study
27	Mayfitriana et al., 2022 [27]	Growth Stunting Prevention in Indonesia: Dentist Knowledge and Perception	Analytic descriptive study
28	Maryati et al., 2022 [28]	The Effect of Interactive Education Program in Preventing Stunting for Mothers with Children under 5 Years of Age in Indonesia: A Randomized Controlled Trial	Randomized controlled trial
29	Mediani et al., 2022 [29]	Factors Affecting the Knowledge and Motivation of Health Cadres in Stunting Prevention Among Children in Indonesia	Cross-sectional study
30	Mindarsih et al., 2023 [30]	Empowerment Model of Pregnant Women in Stunting Prevention Efforts	Descriptive explanatory survey
31	Muliadi et al., 2023 [31]	The coverage of indicators of sensitive and specific intervention programs and prevalence of stunting under-five children: A cross-sectional study in Aceh Province, Indonesia	Cross-sectional study
32	Mulyani et al., 2022 [32]	Synergy from Village, Integrated Healthcare Center, and Early Childhood Education in Stunting Prevention (Case Study)	Case study
33	Mutiarasari et al., 2021 [33]	A Determinant Analysis of Stunting Prevalence on Under5-Year-Old Children to Establish Stunting Management Policy	Case-control study
34	Nawan et al., 2020 [34]	The Relationships of Environmental Sanitation with Stunting among Toddlers Aged 12–36 Months in Bogor Regency, West Java Province, Indonesia	Cross-sectional study
35	Nazri et al., 2016 [35]	Factors influencing mother’s participation in Posyandu for improving nutritional status of children under-five in Aceh Utara district, Aceh province, Indonesia	Cross-sectional study
36	Neherta et al., 2021 [36]	Primary Prevention of Neglect in Children through Health Education for Adolescent Girls in West Sumatra, Indonesia	Pseudo-experimental
37	Nizaruddin et al., 2023 [37]	The Effect of Sanitation on Stunting Prevalence in Indonesia	Cross-sectional study
38	Novitasari et al., 2020 [38]	Maternal feeding practice and its relationship with stunting in children	Cross-sectional study
39	Oktaviana et al., 2022 [39]	Effectiveness of health education and infant therapeutic group therapy on baby aged 0–6 months to prevent stunting risk factors: Maternal depression	Quasi-experimental study
40	Permatasari et al., 2019 [40]	Capacity Building in Health Worker as an Alternative Solution to Solve Stunting Problem	Quantitative and qualitative study
41	Prastiwi et al., 2020 [41]	The Effect of Health Officer Role to the Program of Stunting Prevention on First 1000 Days of Life in Indonesia	Analytic descriptive study
42	Ramli et al., 2009 [42]	Prevalence and risk factors for stunting and severe stunting among under-fives in North Maluku province of Indonesia	Cross-sectional study
43	Rizal et al., 2022 [43]	Stunting Prevention Program of West Sumbawa Regency Health Office: A Qualitative Study in West Nusa Tenggara, Indonesia	Qualitative study
44	Saadah et al., 2022 [44]	Mother Empowerment Model in Stunting Prevention and Intervention through Stunting Early Detection Training	Cross-sectional study and quasi-experimental study
45	Sinaga et al., 2021 [45]	A Qualitative Study of the Effect of Community Participation on Stunting Prevention Behavior in Pandeglang Regency	Analytic descriptive study
46	Sukmawati et al., 2021 [46]	Family Empowerment through Psychosocial Stimulation Assistance and Child Feeding in Increasing Nutrition Intake and Body Weight of Children 2–3 Years Old to Prevent Stunting	Quasi-experimental
47	Suratri et al., 2023 [47]	Risk Factors for Stunting among Children under Five Years in the Province of East Nusa Tenggara (NTT), Indonesia	Cross-sectional study
48	Suryani et al., 2022 [48]	Determinants of Feeding Patterns with Stunting in Children in the Coastal Area of Bengkulu City	Cross-sectional study
49	Syafrawati et al., 2022 [49]	Factors driving and inhibiting stunting reduction acceleration programs at district level: A qualitative study in West Sumatra	Qualitative study design and a case study
50	Susanto et al.,2021 [50]	Prevalence of malnutrition and stunting among under-five children: A cross-sectional study family of quality of life in agricultural areas of Indonesia	Cross-sectional study
51	Titaley et al., 2019 [51]	Determinants of the Stunting of Children Under Two Years Old in Indonesia: A Multilevel Analysis of the 2013 Indonesia Basic Health Survey	Cross-sectional study
52	Widhiastuti et al., 2021 [52]	Reconstruction of Prevention and Handling of Stunting Policy in Public Health Center	Qualitative study

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
