# Peer review of "The Importance of Interprofessional Collaboration (IPC) Guidelines in Stunting Management in Indonesia: A Systematic Review"

_healthcare, 2024, doi:10.3390/healthcare12222226_

Round 1

Reviewer 1 Report

Comments and Suggestions for Authors

I have included my comments in the file attached

Comments on the Quality of English Language

There is need for editing of the english as a number of grammatical errors were cited

Author Response

Dear Reviewers,

We sincerely appreciate your comprehensive review and thoughtful comments on our manuscript. We have carefully addressed all your comments and made the following revisions:

  1. The discrepancy in the number of included articles has been rectified. The PRISMA flowchart now correctly reflects 52 articles, consistent with our systematic review.
  2. All sentences have been revised to comply with proper capitalization conventions in academic writing.
  3. A carefully worded acknowledgment regarding language limitations has been added to the limitations section, emphasizing our methodological considerations while recognizing the potential value of non-English literature.
  4. The sentence about prevention effectiveness has been revised to read
  5. The prevalence statistics have been enhanced with clear geographical context and specified as being based on WHO child growth standards, improving the clarity and standardization of the reported measurements.
  6. All identified grammatical concerns have been addressed to improve readability and precision.
  7. Detailed explanations of therapeutic formulas (F-75 and F-100) have been incorporated to enhance understanding for all readers.
  8. A thorough review of the manuscript has been conducted to address all linguistic and clarity issues.
  9. Regarding the comment on analysis methodology: We have substantially expanded section 2.4 "Data synthesis and analysis" to provide a more detailed description of our analytical approach.
  10. In the Discussion section, we have added substantive paragraphs that distill major lessons learned regarding interprofessional collaboration frameworks, community engagement strategies, and implementation requirements.
  11. The Author Contributions section has been restructured to provide explicit delineation of each author's specific role in the study.

All revisions have been highlighted in yellow in the main document for easy reference and review.

We hope these revisions have adequately addressed your concerns and improved the quality of our manuscript. Should you require any further clarification or additional modifications, we would be pleased to address them.

Thank you for your valuable guidance in improving this work.

Best regards,

The Authors

Reviewer 2 Report

Comments and Suggestions for Authors

Dear authors,

This qualitative systematic review on healthcare approaches and challenges in Indonesia in addressing infant stunting gives comprehensive overview of various stages in healthcare regarding this issues, identifies roles of different stakeholders in these processes, highlighting what still needs to be done. 

A bit more on how the analysis was performed could improve the soundness of the study. Also try to summaries some major lessons learned as a guidelines and take away message.

I marked where I saw that language needs editing.

Comments on the Quality of English Language

Please improve highlighted parts of the text.

Author Response

Dear Reviewers,

We sincerely appreciate your comprehensive review and thoughtful comments on our manuscript. We have carefully addressed all your comments and made the following revisions:

  1. The discrepancy in the number of included articles has been rectified. The PRISMA flowchart now correctly reflects 52 articles, consistent with our systematic review.
  2. All sentences have been revised to comply with proper capitalization conventions in academic writing.
  3. A carefully worded acknowledgment regarding language limitations has been added to the limitations section, emphasizing our methodological considerations while recognizing the potential value of non-English literature.
  4. The sentence about prevention effectiveness has been revised to read
  5. The prevalence statistics have been enhanced with clear geographical context and specified as being based on WHO child growth standards, improving the clarity and standardization of the reported measurements.
  6. All identified grammatical concerns have been addressed to improve readability and precision.
  7. Detailed explanations of therapeutic formulas (F-75 and F-100) have been incorporated to enhance understanding for all readers.
  8. A thorough review of the manuscript has been conducted to address all linguistic and clarity issues.
  9. Regarding the comment on analysis methodology: We have substantially expanded section 2.4 "Data synthesis and analysis" to provide a more detailed description of our analytical approach.
  10. In the Discussion section, we have added substantive paragraphs that distill major lessons learned regarding interprofessional collaboration frameworks, community engagement strategies, and implementation requirements.
  11. The Author Contributions section has been restructured to provide explicit delineation of each author's specific role in the study.

All revisions have been highlighted in yellow in the main document for easy reference and review.

We hope these revisions have adequately addressed your concerns and improved the quality of our manuscript. Should you require any further clarification or additional modifications, we would be pleased to address them.

Best regards,

The Authors

Reviewer 3 Report

Comments and Suggestions for Authors

I am grateful for the opportunity to review this article and extend my congratulations to the authors on their work. I find the article interesting, relevant, and current. It is well structured with relevant data.

My primary concern pertains to the article's suitability for a broader readership. Therefore, it is important to explain in the introduction the rationale for publishing this manuscript, which deals with stunting management in Indonesia, in a global journal. It would be beneficial to ascertain whether this paper can contribute to the approach to this problem in other countries or different geographical areas.

The discussion is pertinent to the data presented in the articles. The references are adequate and up to date.

In addition, I have a few minor comments, namely:

·      The PRISMA flowchart identifies 54 articles included in the review; however, the authors identify 52 in the text. Please clarify this discrepancy.

·      Line 225 – Start sentence with a capital letter.

·      The decision to include only articles written in English represents a limitation in itself. Moreover, it is possible that the exclusion of non-indexed articles, which pertains specifically to a single country, may have resulted in the omission of significant studies.

Author Response

(The authors gave the same response as above.)
